# Unsupervised Binary Code Translation with Application to Code Similarity Detection and Vulnerability Discovery

**Iftakhar Ahmad**
University of South Carolina
iahmad@email.sc.edu

**Lannan Luo** ✉
George Mason University
lluo4@gmu.edu

## Abstract

Binary code analysis has immense importance in the research domain of software security. Today, software is very often compiled for various Instruction Set Architectures (ISAs). As a result, cross-architecture binary code analysis has become an emerging problem. Recently, deep learning-based binary analysis has shown promising success. It is widely known that training a deep learning model requires a massive amount of data. However, for some low-resource ISAs, an adequate amount of data is hard to find, preventing deep learning from being widely adopted for binary analysis. To *overcome the data scarcity problem* and *facilitate cross-architecture binary code analysis*, we propose to apply the ideas and techniques in Neural Machine Translation (NMT) to binary code analysis. Our insight is that a binary, after disassembly, is represented in some assembly language. Given a binary in a low-resource ISA, we *translate* it to a binary in a high-resource ISA (e.g., x86). Then we can use a model that has been trained on the high-resource ISA to test the translated binary. We have implemented the model called UNSU-PERBINTRANS, and conducted experiments to evaluate its performance. Specifically, we conducted two downstream tasks, including *code similarity detection* and *vulnerability discovery*. In both tasks, we achieved high accuracies.

## 1 Introduction

In software security research, binary code analysis plays a significant role. It can play a vital role in various tasks, such as vulnerability discovery, code plagiarism detection, and security auditing of software, etc., without accessing the source code. Today, software is very often compiled for various Instruction Set Architectures (ISAs). For example, IoT vendors often use the same code base to compile firmware for different devices that operate on varying ISAs (e.g., x86 and ARM), which causes a single vulnerability at source-code level to spread across binaries of diverse devices. As a result, *cross-architecture binary code analysis* has become an emerging problem (Pewny et al., 2015; Eschweiler et al., 2016; Feng et al., 2016; Xu et al., 2017; Zuo et al., 2018). Analysis of binaries across ISAs, however, is non-trivial: such binaries differ greatly in instruction sets; calling conventions; general- and special-purpose CPU register usages; and memory addressing modes.

Recently, we have witnessed a surge in research efforts that leverage deep learning to tackle various binary code analysis tasks, including code clone detection (Luo et al., 2017a; Zuo et al., 2018), malware classification (Raff et al., 2018; Cakir and Dogdu, 2018), vulnerability discovery (Lin et al., 2019; Wu et al., 2017), and function prototype inference (Chua et al., 2017). Deep learning has demonstrated its strengths in code analysis, and shown noticeably better performances over traditional program analysis-based methods (Song et al., 2013; BAP, 2011; Luo et al., 2014; Wang et al., 2009b) in terms of both accuracy and scalability.

It is widely known that training a deep learning model usually requires a massive amount of data. As a result, most deep learning-based binary code analysis methods have been dedicated to a *high-resource* ISA, such as x86, where large-scale labeled datasets exist for training their models. But for many other ISAs (e.g., ARM, PowerPC), there are few or even no labeled datasets, resulting in negligent focus on these *low-resource* ISAs. Moreover, it is labor-intensive and time-consuming to collect data samples and manually label them to build datasets for such low-resource ISAs. In this work, we aim to *overcome the challenge of data scarcity in low-resource ISAs* and *facilitate cross-architecture binary code analysis*.

**Our Insight.** Given a pair of binaries in different ISAs, they, after being disassembled, are represented in two sequences of instructions in two assembly languages. In light of this insight, we

propose to learn from Neural Machine Translation (NMT), an area that focuses on translating texts from one human language into another using machine/deep learning techniques (Sutskever et al., 2014). There are two types of NMT systems. One is supervised and requires a large parallel corpus, which poses a major practical problem. Another is unsupervised, which removes the need for parallel data. As unsupervised NMT has the advantage of requiring no cross-lingual signals, we adopt unsupervised NMT for binary code analysis.

**Our Approach.** Drawing inspiration from the unsupervised neural machine translation model Undreamt (Artetxe et al., 2018b), we design an unsupervised binary code translation model called UNSUPERBINTRANS. UNSUPERBINTRANS is trained in a completely unsupervised manner, relying solely on mono-architecture corpora. Once trained, UNSUPERBINTRANS can translate binaries from a low-resource ISA to a high-resource ISA (e.g., x86). Consequently, when dealing with a binary in a low-resource ISA, translating it into a high-resource ISA enables us to use a model trained on the high-resource ISA to test the translated binary code. UNSUPERBINTRANS can be applied to a variety of applications, such as vulnerability discovery and code similarity detection.

We have implemented UNSUPERBINTRANS [1], and evaluated its performance. Specifically, we conduct two critical binary analysis tasks, including *code similarity detection* and *vulnerability discovery*. In both tasks, we achieve high accuracies across different ISAs. For example, a model trained on x86 can successfully detect *all* vulnerable functions in ARM (even the model has not been exposed to any vulnerable functions in ARM during its training), indicating the exceptional translation capabilities of UNSUPERBINTRANS.

Below we summarize our contributions:

- We propose a novel *unsupervised* method for translating binary code from one ISA to another, such that the translated binary shares similar semantics as the original code.

- We have implemented the model UNSU-PERBINTRANS and conducted evaluations on two important binary analysis tasks. The results show that UNSUPERBINTRANS can successfully capture code semantics and effectively translate binaries across ISAs.

---

[1] https://github.com/lannan/UnsuperBinTrans

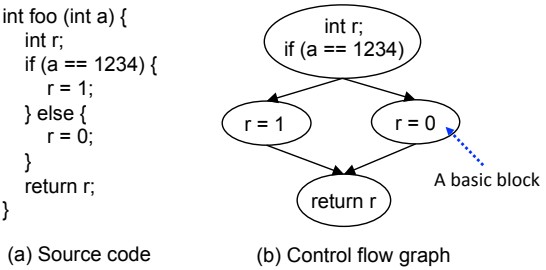

```
int foo (int a) {
    int r;
    if (a == 1234) {
        r = 1;
    } else {
        r = 0;
    }
    return r;
}
```

(a) Source code  (b) Control flow graph

Figure 1: Control flow graph and basic block.

- This work proposes a new research direction in binary code analysis, especially for low-resource ISAs. Based on it, for a downstream binary analysis task, we only need to train a model on a high-resource ISA and transfer it to analyze binaries in other ISAs by translating such binaries to the high-resource ISA. Through this, we can resolve the data scarcity issue in low-resource ISAs.

## 2 Background

A control flow graph (CFG) is the graphical representation of control flow or computation during the execution of programs or applications. A CFG is a directed graph in which each node represents a basic block and each edge represents the flow of control between blocks. A basic block is a sequence of consecutive statements in which the flow of control enters at the beginning and leaves at the end without halt or branching except at the end.

As shown in Figure 1(a), given a piece of source code, its corresponding CFG is shown in Figure 1(b), where each node is a basic block. Similarly, we can also generate the corresponding CFG for a piece of binary code. We here use the source code as an example for simplicity.

## 3 Motivation and Model Design

This section first discusses the motivation behind our proposal for unsupervised binary code translation, and then presents the model design.

### 3.1 Motivation

Consider the vulnerability discovery task, a longstanding problem of binary code analysis, as an example, binaries in different ISAs are analyzed in detail to find vulnerabilities. Nowadays, applying machine/deep learning to binary analysis has drawn great attention due to the exceptional performance. However, it usually requires a large amount of data for training. Unfortunately, many a time it

is difficult to collect such a large amount of training data, especially for low-resource ISAs.

There are some ISAs that are commonly used (e.g., x86) and it becomes easy to collect large training data for such ISAs. It would be a great advantage if the availability of such large training data of high-resource ISAs could facilitate the automated analysis of binaries in low-resource ISAs, where sufficient training data is not available.

For example, suppose there is a large training dataset for the ISA **X**, however, we need to analyze a binary *b* in the ISA **Y**, where the available training data is insufficient. As a result, it is difficult to train a model for analyzing *b* in **Y**. In order to resolve the problem, our idea is to translate *b* from the ISA **Y** to **X**, resulting in a translated binary *b'* in **X**. Then, we can leverage the large training dataset of **X** to train a model, which can be used to perform prediction on the translated version *b'* in **X**.

## 3.2 Learning Cross-Architecture Instruction Embeddings

A binary, after disassembled, is represented as a sequence of instructions. An instruction includes an opcode (specifying the operation to be performed) and zero or more operands (specifying registers, memory locations, or literal data). For example, `mov eax, ebx` is an instruction where `mov` is an opcode and both `eax` and `ebx` are operands.[2] In NMT, words are usually converted into word embeddings to facilitate further processing. Since we regard instructions as words, similarly we represent instructions as *instruction embeddings*.

**Challenge.** In NLP, if a trained word embedding model is used to convert a word that has never appeared during training, the word is called an *out-of-vocabulary* (OOV) word and the embedding generation for such words will fail. This is a well-known problem in NLP, and it exacerbates significantly in our case, as constants, address offsets, and labels are frequently used in instructions. How to deal with the OOV problem is a challenge.

**Solution: Preprocessing Instructions.** To resolve the OOV problem, we propose to preprocess the instructions using the following rules: (1) Constants up to 4 digits are preserved because they usually contain useful semantic information; constants more than 4 digits are replaced with `<CONST>`. (2) Memory addresses are replaced with `<ADDR>`.

---

[2]Assembly code in this paper adopts the Intel syntax, i.e., `op dst, src(s)`.

(3) Other symbols are replaced with `<TAG>`. Take the code snippets below as an example, where the left one shows the original assembly code and the right one is the preprocessed code.

```
MOV EDX, 11E1H          MOV EDX, 11E1H
MOV ECX, 0FFFFFFFFH     MOV ECX, <CONST>
JLE LOC_9BA3B           JLE LOC_<TAG>
CALL CRYPTO_FREE        CALL CRYPTO_FREE
MOV RCX, CS:GLIBC_2_5   MOV RCX, CS:<ADDR>
MOV [RSP+VAR_58], RDX   MOV [RSP+], RDX
```

**Building Training Datasets.** As we regard instructions as words and basic blocks as sentences, we use a large number of basic blocks to train the instruction embeddings. Specifically, for a given ISA **X**, we first collect various opensource programs, and compile them in **X**. Note that given the wide availability of opensource code, this requires little effort. After that, we use IDA Pro (The IDA Pro Disassembler and Debugger) to disassemble all the binaries to generate the assembly code, from which we can extract the basic blocks. We use all the basic blocks (after preprocessing) to build a training dataset for this ISA.

**Learning Process.** Below we define two terms, *mono-architecture instruction embeddings* (MAIE) and *cross-architecture instruction embeddings* (CAIE).

**Definition 1** (*Mono-Architecture Instruction Embeddings*) *Mono-architecture instruction embeddings (MAIE) are architecture-specific, where similar instructions in the same ISA have close embeddings. MAIE of different ISAs are projected into different vector spaces.*

**Definition 2** (*Cross-Architecture Instruction Embeddings*) *Cross-architecture instruction embeddings (CAIE) are architecture-agnostic, where similar instructions, regardless of their ISAs, have close embeddings. CAIE are projected into the same vector space.*

We adopt `fastText` (Bojanowski et al., 2017) to learn MAIE for an ISA. The result is an embedding matrix $\mathbf{X} \in \mathbb{R}^{V \times d}$, where $V$ is the number of unique instructions in the vocabulary, and $d$ is the dimensionality of the instruction embeddings. The $i$th row $\mathbf{X}_{i*}$ is the embedding of the $i$th instruction in the vocabulary.

Given two ISAs, where one is a high-resource ISA and another a low-resource ISA, we follow the aforementioned steps to learn MAIE for each ISA. After that, we map MAIE of the low-resource

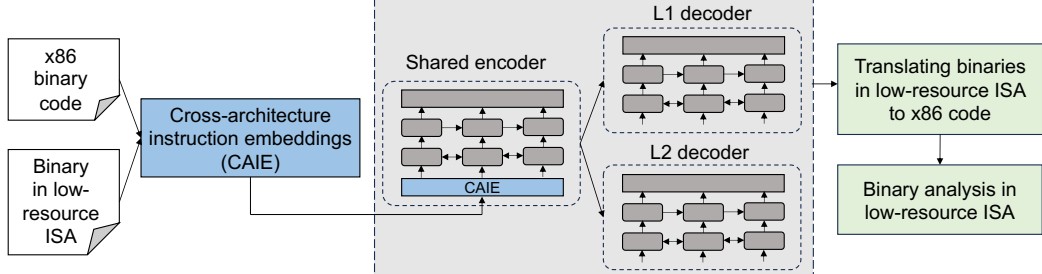

Figure 2: Architecture of UNSUPERBINTRANS.

ISA into the vector space of the high-resource ISA by adopting the unsupervised learning technique vecmap (Artetxe et al., 2018a). Note that other unsupervised learning methods for generating cross-lingual word embeddings (Conneau et al., 2017; Ruder et al., 2019) also work.

### 3.3 Translating Basic Blocks Across ISAs

Drawing inspiration from the unsupervised neural machine translation model Undreamt (Artetxe et al., 2018b), we design an unsupervised binary code translation model, named UNSUPERBIN-TRANS, to translate a piece of binary code (i.e., a basic block) from one ISA to another ISA. The model architecture of UNSUPERBINTRANS is shown in Figure 2.

First, CAIE are generated for two ISAs, where one is a high-resource ISA (x86) and another a low-resource ISA (see Section 3.2). Then, we train UNSUPERBINTRANS (the grey dashed box; the detailed process is discussed below). The training only uses mono-architecture datasets and CAIE generated from the mono-architecture datasets. The mono-architecture dataset for each ISA contains a large number of basic blocks from this ISA. Finally, the trained model can translate a basic block from the low-resource ISA to x86, and the translated basic block shares similar semantics as the original one. Through the translation of binaries from a low-resource ISA to x86, we can use a model trained on x86 to analyze the translated code. This capability allows UNSUPERBINTRANS to address the data scarcity issue in low-resource ISAs. UNSUPERBINTRANS can be applied to various binary analysis tasks, such as code similarity detection and vulnerability discovery.

As UNSUPERBINTRANS is designed based on Undreamt, below we briefly discuss how Undreamt works. Please refer to (Artetxe et al., 2018b) for more details. Undreamt alleviates the major limitations of NMT, i.e., the requirement of very large

parallel text corpora. Undreamt achieves very good results for translation due to the three main properties of its architecture. These properties are *dual structure*, *shared encoder*, and *fixed embeddings in the encoder*. First, the dual structure enables bidirectional translation capabilities to be achieved through training. Second, the shared encoder produces a language-independent representation of the input text corpus. Then the corresponding decoders for a particular language can transform that representation into the required language. Finally, with the help of fixed embeddings in the encoder, Undreamt can easily learn how to compose word-level representations of a language to build representations for bigger phrases. These embeddings are pre-trained and kept fixed throughout the training which makes the training procedure simpler.

The training procedure consists of two main underlying components. *Denoising* and *On-the-fly backtranslation*. Although the dual property of Undreamt helps it gain preliminary translation knowledge, eventually such procedure results in a simple copying task where the trained model may result in making word-by-word replacements of the source and target languages. Therefore, to improve the translation capability of the trained model, Undreamt injects noise into input data by adding some random swaps of contiguous words. Then the model is asked to recover the original input text by denoising the corrupted data. This process of adding and removing the noise helps the model to learn the actual relationship between words in the sentences for the given languages.

We implement UNSUPERBINTRANS upon Undreamt. During the testing phase, when provided with a basic block from one ISA, we first represent it as a sequence of CAIE, which is fed to UNSUPERBINTRANS. UNSUPERBINTRANS then translates it to a basic block in another ISA, which shares similar semantics as the source basic block.

Table 1: Statistics of datasets.

| Opt. Level | ISA | # of Functions | # of Unique Instructions | Total # of Instructions |
|---|---|---|---|---|
| O0 | ARM | 80,065 | 100,515 | 6,669,266 |
| | x86 | 71,608 | 94,121 | 6,555,483 |
| O1 | ARM | 83,184 | 102,388 | 6,864,017 |
| | x86 | 70,350 | 89,893 | 6,587,172 |
| O2 | ARM | 74,173 | 111,564 | 7,408,810 |
| | x86 | 70,678 | 91,139 | 6,626,772 |
| O3 | ARM | 74,186 | 111,587 | 7,317,972 |
| | x86 | 70,329 | 89,932 | 6,619,398 |

## 4 Evaluation

We conducted experiments to evaluate the translation performance of UNSUPERBINTRANS. We first describe the experimental settings, and then conduct quantitative analysis by computing the BLEU scores (Bilingual Evaluation Understudy) (Papineni et al., 2002). After that, we conduct two downstream tasks, including code similarity comparison and vulnerability discovery. Finally, we evaluate the efficiency.

### 4.1 Experimental Settings

UNSUPERBINTRANS contains three components, including fastText (Bojanowski et al., 2017), vecmap (Artetxe et al., 2018a), and Undreamt (Artetxe et al., 2018b). The experiments were conducted on a computer with Ubuntu 20.04, a 64-bit 2.50 GHz 11th Gen Intel® Core(TM) i7-11700 CPU with 8 cores, an Nvidia GeForce RTX 3080 GPU, 64 GB RAM, and 2 TB HD. We set the embedding dimension to 200 and use the default settings of vecmap and Undreamt: the hyperparameters, such as learning rate, epoch, and batch size, etc., are set to their default values.

**Datasets.** We consider two ISAs, x86 and ARM. We select a set of widely used open-source packages, include binutils, coreutils, findutils, inetutils, and openssl, to build two mono-architecture datasets (one for x86 and another for ARM). Specifically, for each program, we compile it into binaries in x86 and ARM. We consider four optimization levels, O0, O1, O2, and O3, when generating binaries. As a result, for each program, we obtain four binaries for each ISA, each of which is for one optimization level.

We next disassemble these binaries using IDA Pro and extract basic blocks. Each instruction in basic blocks is then preprocessed. Finally, we build *four* mono-architecture datasets for x86 and another *four* for ARM. Each mono-architecture

dataset contains a set of preprocessed basic blocks corresponding to one ISA and one optimization level. Table 1 shows the statistics of our datasets, including the number of functions, the number of unique instructions (i.e., vocabulary size), and the total number of instructions for each optimization level in terms of the two ISAs, x86 and ARM.

**Training UNSUPERBINTRANS.** Given two datasets corresponding to two different ISAs and the same optimization level, we first use fastText to generate MAIE for each ISA, which are then mapped to a shared space to generate CAIE by vecmap. After that, we train UNSUPERBINTRANS using the two datasets and CAIE to translate basic blocks from one ISA to another. Note that UNSUPERBINTRANS is based on unsupervised learning; we do not need to determine which basic block in one ISA is similar to a basic block in another ISA.

### 4.2 Quantitative and Qualitative Analysis

We first calculate the BLEU score to measure the quality of our translation. A BLEU score is represented by a number between zero and one. If the score is close to zero, the translation quality is poor. On the other hand, the translation quality is good when the score is close to one.

For each x86 function, we first find its similar ARM function: following the dataset building method in InnerEye (Zuo et al., 2018), we consider two functions similar if they are compiled from the same piece of source code, and dissimilar if their source code is rather different. As a result, for each ARM function, denoted as $S$, upon identifying its similar x86 function $R$, we include this similar pair into our dataset. For each ARM function in the dataset, we use UNSUPERBINTRANS to translate each of its basic blocks into x86, resulting in a translated function in x86, denoted as $T$. After that, we compute the BLEU score between $T$ and $R$. Finally, we compute the average BLEU score for all functions in the dataset. We obtain 0.76, 0.77, 0.77, 0.76 BLEU scores, for the four different optimization levels, respectively, when using fastText for instruction embedding generation. Moreover, when we use word2vec, we obtain 0.77, 0.77, 0.77, 0.76 BLEU scores, for the four different optimization levels, respectively. Thus, our model achieves good translation.

Table 2 shows three randomly selected examples. Due to space limits, we use basic blocks as examples. In each example, 1) the *source ARM* is the

Table 2: Three examples of assembly code translations.

| | | |
|---|---|---|
| **1** | Source ARM | LDR R0, [R7, 0X10+<ADDR>]; MOV R1, <TAG>; BL STARTSWITH_0; MOV R3, R0; CMP R3, #0; BEQ LOC_<TAG> |
| | Reference x86 | MOV RAX, [RBP+NAME]; LEA RSI, <TAG>; MOV RDI, RAX; CALL STARTSWITH_0; TEST AL, AL; JZ SHORT LOC_<TAG> |
| | Translated x86 | MOV RAX, [RBP+NAME]; LEA RSI, <TAG>; MOV RDI, RAX; CALL _STRCMP; TEST EAX, EAX; JZ SHORT LOC_<TAG> |
| **2** | Source ARM | LDR R3, [R7, 0X18+H]; LDRB.W R3, [R3, #0X37]; AND.W R3, R3, #2; UXTB R3, R3; CMP R3, #0; BNE LOC_<TAG> |
| | Reference x86 | MOV RAX, [RBP+H]; MOVZX EAX, <BYTE_PTR>[RAX+6BH]; AND EAX, 2; TEST AL, AL; JNZ SHORT LOC_<TAG> |
| | Translated x86 | MOV RAX, [RBP+H]; MOVZX EAX, <BYTE_PTR>[RAX+6BH]; AND EAX, 2; TEST AL, AL; JNZ SHORT LOC_<TAG> |
| **3** | Source ARM | LDR R3, [R7, 0X18+INFO]; LDRB R3, [R3, #6]; AND.W R3, R3, #4; UXTB R3, R3; CMP R3, #0; BEQ LOC_<TAG> |
| | Reference x86 | MOV RAX, [RBP+H]; MOVZX EAX, <_PTR>[RAX+6CH]; AND EAX, 20H; TEST AL, AL; JNZ SHORT LOC_<TAG> |
| | Translated x86 | MOV RAX, [RBP+INFO]; MOVZX EAX, <BYTE_PTR>[RAX+6]; AND EAX, 4; TEST AL, AL; JZ SHORT LOC_<TAG> |

original ARM basic block; 2) the *reference x86* is the x86 basic block that is similar to the original ARM basic block; and 3) the *translated x86* is the x86 basic block that is translated from the original ARM basic block by our model. By comparing the translated x86 block with the reference x86 block, we can see that UNSUPERBINTRANS (1) successfully predicts all opcodes (i.e., the opcodes in the translated x86 and those in the reference x86 block are the same), and (2) a small number of operands are not predicted correctly but reasonable. For example, in the first example, the fifth instruction in the reference x86 block is TEST AL, AL, while the predicted instruction in the translated x86 block is TEST, EAX, EAX. Opcodes, which determines the operation to be performed, captures more semantics compared to operands. Moreover, the random allocation of registers across different optimization levels diminishes their significance. We thus can conclude that UNSUPERBINTRANS consistently attains exceptionally high-quality translations. This results in superior performance in various downstream tasks, as detailed in the following sections.

### 4.3 Downstream Applications

We consider two downstream tasks as follows.

**Function Similarity Comparison.** Given two binary functions, $f_1$ and $f_2$, in two different ISAs, $\mathbf{X}$ and $\mathbf{Y}$, our goal is to measure their similarity. If they are similar, they may copy from each other (i.e., stem from the same piece of source code). To achieve this, we translate each basic block of $f_1$ in

the ISA $\mathbf{X}$ to a basic block in the ISA $\mathbf{Y}$, resulting in a function $f_1'$ in $\mathbf{Y}$. After that, we have two binary functions, $f_1'$ and $f_2$, within the same ISA $\mathbf{Y}$.

For each function, we generate its function embedding, which is computed as the weighted sum of the CAIE of all instructions in this function. The weight of each CAIE is calculated as the term-frequency (TF). We then compute the cosine similarity between their function embeddings to measure their similarity.

**Vulnerability Discovery.** We consider three vulnerabilities: CVE-2014-0160 (or Heartbleed bug) (Banks, 2015), CVE-2014-3508, and CVE-2015-1791. We follow a similar way as above to compute the functions embeddings for the vulnerable functions and benign functions. We use Support Vector Machine (SVM), a widely used machine learning model to detect vulnerabilities.

Since there is usually only one vulnerable function (positive sample) for training SVM, the dataset becomes extremely imbalanced. To handle the imbalance issue, Random Oversampling (ROS) (Moreo et al., 2016) and Synthetic Minority Over-sampling Technique (SMOTE) (Chawla et al., 2002) are adopted. For each vulnerability case, we first train an SVM model using the vulnerable functions and benign functions in x86. We then use the trained model to detect the vulnerability in ARM functions. During detection, we first translate each basic block in the ARM function to a basic block in x86 and use the SVM model (that has been trained on x86) to test the translated function.

Table 3: Accuracy of the function similarity task.

| Opt. Level | Tool for instruction embedding generation | |
| --- | --- | --- |
| | word2vec | fastText |
| O0 | 78.09% | 81.24% |
| O1 | 83.33% | 91.11% |
| O2 | 87.05% | 95.30% |
| O3 | 86.76% | 94.97% |

## 4.4 Function Similarity Comparison

For each optimization level, we first randomly select $7,418$ x86 functions. For each selected x86 function, we search for its similar and dissimilar function in ARM. We consider two functions similar if they are compiled from the same piece of source code, and dissimilar if their source code is rather different. We assign the label "1" to each similar function pair, and the label "0" to each dissimilar function pair.

We next generate function embeddings for each function. In NLP, many methods exist for composing word embeddings to sentence/document embeddings. We choose the following way. We calculate the term-frequency (TF) for each instruction in a function and then multiply it by the corresponding CAIE of that instruction. We next add all the weighted CAIE and use the sum as the function embedding. Note that for an ARM function, we first translate it to an x86 function using UNSUPERBIN-TRANS, and then compute its function embedding using the method discussed before. To measure the function similarity, we calculate the cosine similarity between their function embeddings and then calculate the accuracy.

We perform the experiment for all the optimization levels. Table 3 shows the results. As UN-SUPERBINTRANS needs to first generate MAIE (see Section 3.2) and then perform the translation, we also explore different tools for generating MAIE and assess which one yields better performance. Specifically, we use fastTtext and word2vec. The results demonstrate that we achieve reasonable accuracies for each optimization level. Moreover, fastText leads to better performance.

Note that in this task, we use a weighted sum of instruction embeddings to generate function embeddings. As there are many advanced methods for composing word embeddings, including those using machine/deep learning models (Lin et al., 2017; Kenter et al., 2016; Xing et al., 2018), we expect that by adopting these advanced methods, we can achieve even higher accuracies.

## 4.5 Vulnerability Discovery

We consider three case studies as follows.

**Case Study I.** CVE-2014-0160 is the *Heartbleed* vulnerability in OpenSSL (Banks, 2015). It allows remote attackers to get sensitive information via accessing process stack memory using manually crafted data packets which would trigger a buffer overread. We obtain the vulnerable binary function in x86 and ARM from OpenSSL v1.0.1d.

**Case Study II.** For CVE-2014-3508, context-dependent attackers are able to obtain sensitive information from process stack memory via reading output from some specific functions. The vulnerable function is collected from OpenSSL v1.0.1d.

**Case Study III.** For CVE-2015-1791, a race condition occurs when the vulnerable function is used for a multi-threaded client. A remote attacker can create a denial of service attack (double free and application crash) when attempting to reuse a ticket that had been obtained earlier. The vulnerable function is also collected from OpenSSL v1.0.1d.

**Our Results.** For each case study, we select 9,999 benign functions. However, we only have one vulnerable sample. To address this unbalanced sample issue, we use ROS (Moreo et al., 2016) and SMOTE (Chawla et al., 2002). We set the parameters *k_neighbors* to 2 and *sampling_strategy* to 0.002. We try different values of *sampling_strategy*, including 0.5, 0.25, and 0.02, etc., and find that with the value of 0.002, we obtain enough oversampling data and yield good results. As we set *k_neighbors* to 2, we duplicate the vulnerable function three times, so that the initial minority sample size is more than the value of *k_neighbors*.

We then compute the function embedding for each function. We use the function embeddings of x86 vulnerable functions and benign functions to train an SVM model. Then, the trained SVM model is transferred to test the translated code of each ARM function, *without any modification*. In this experiment, we use fastText to generate MAIE as it gives better results than word2vec.

***Baseline Method.*** Our work aims to address the data scarcity issue in low-resource ISAs. To achieve this, we introduce UNSUPERBINTRANS, which can translate binaries from a low-resource ISA to a high-resource ISA. By applying UN-SUPERBINTRANS, we can use the SVM model trained on x86 to test a binary in ARM by translating the ARM binary to x86.

Table 4: Performance of the vulnerability discovery task (%).

| Opt. Level | Case Study | True Positive Rate | | False Positive Rate | | Precision | | F1-score | |
| --- | --- | --- | --- | --- | --- | --- | --- | --- | --- |
| | | Baseline | **Ours** | Baseline | **Ours** | Baseline | **Ours** | Baseline | **Ours** |
| | I | 1.00 | 1.00 | 0.00 | 0.0001 | 1.00 | 0.95 | 1.00 | 0.98 |
| O0 | II | 1.00 | 1.00 | 0.0002 | 0.0001 | 0.83 | 0.95 | 0.91 | 1.00 |
| | III | 1.00 | 1.00 | 0.00 | 0.0001 | 1.00 | 0.95 | 1.00 | 0.98 |
| | I | 1.00 | 1.00 | 0.00 | 0.0001 | 1.00 | 0.95 | 1.00 | 0.98 |
| O1 | II | 1.00 | 1.00 | 0.0003 | 0.0002 | 0.77 | 0.83 | 0.87 | 0.91 |
| | III | 1.00 | 1.00 | 0.00 | 0.0002 | 1.00 | 0.83 | 1.00 | 0.91 |
| | I | 1.00 | 1.00 | 0.00 | 0.00 | 1.00 | 1.00 | 1.00 | 1.00 |
| O2 | II | 1.00 | 1.00 | 0.0001 | 0.00 | 0.91 | 1.00 | 0.95 | 1.00 |
| | III | 1.00 | 1.00 | 0.00 | 0.00 | 1.00 | 1.00 | 1.00 | 1.00 |
| | I | 1.00 | 1.00 | 0.00 | 0.00 | 1.00 | 1.00 | 1.00 | 1.00 |
| O3 | II | 1.00 | 1.00 | 0.0002 | 0.00 | 0.83 | 1.00 | 0.91 | 1.00 |
| | III | 1.00 | 1.00 | 0.00 | 0.0001 | 1.00 | 0.95 | 1.00 | 0.98 |

To showcase the effectiveness of the translation, we perform a baseline comparison. The baseline model we consider is an SVM model trained and tested on ARM, *without employing any translation*. As one would expect, the model trained and tested on ARM is likely to outperform a model trained on x86 and tested on the translated code (from ARM to x86). That is, the baseline model represents the best-case scenario. Moreover, *if the performance difference between the baseline and our model is small, it signifies effective translation.*

Table 4 shows the results. We can see that our model consistently demonstrates close proximity in most cases when comparing to the baseline. The performance of UNSUPERBINTRANS across all performance metrics is impressive. For example, UNSUPERBINTRANS has a 100% true positive rate, indicating it identifies all the ARM vulnerable functions accurately. Therefore, we can conclude that UNSUPERBINTRANS has exceptional translation capabilities and can be applied to the vulnerability discovery task with high reliability.

## 4.6 Efficiency

We evaluate the training time of UNSUPERBIN-TRANS, which needs to first generate MAIE using `fastText` (Part I) and then learn CAIE (Part II). After that, we train UNSUPERBINTRANS using CAIE and two mono-architecture datasets (Part III). The total training time is the sum of the three parts. Part I approximately around 20 minutes for MAIE generation of ARM and x86. Part II takes around 15 minutes for CAIE generation. Part III takes approximately 16 hours for the training of each optimization level. Thus, the total training time is around 16.5 hours.

## 5 Related Work

The related work can be divided into: traditional and machine/deep learning based. Each one can be further divided into: mono-architecture and cross-architecture based.

### 5.1 Traditional Code Similarity Comparison

**Mono-Architecture Approaches.** Most traditional approaches work on a *single* ISA. Some analyze source code (Kamiya et al., 2002; Jiang et al., 2007; Luo and Zeng, 2016). Others analyze binary code (Luo, 2020; Zeng et al., 2019b; Luo et al., 2019a; Zeng et al., 2019a, 2018; Luo et al., 2016), e.g., using symbolic execution (Luo et al., 2014, 2021, 2017b), but are expensive. Dynamic approaches include API birthmark (Tamada et al., 2004), system call birthmark (Wang et al., 2009a), instruction birthmark (Tian et al., 2013), and core-value birthmark (Jhi et al., 2011). However, *extending them to other ISAs would be hard and tedious*. Moreover, code coverage is another challenge.

**Cross-Architecture Approaches.** Recent works have applied traditional approaches to the *cross-architecture* scenario (Pewny et al., 2015; Eschweiler et al., 2016; Chandramohan et al., 2016; Feng et al.; David et al., 2018). `Multi-MH` and `Multi-k-MH` (Pewny et al., 2015) are the first two methods for comparing functions across ISAs, but their fuzzing-based basic-block similarity comparison and graph (i.e., CFG) matching-based algorithms are expensive. `discovRE` (Eschweiler et al., 2016) uses pre-filtering to boost CFG-based matching process, but is still expensive and unreliable. `Esh` (David et al., 2016) uses data-flow slices of basic blocks as comparable unit. It uses a SMT solver to verify function similarity, which is unscalable.

## 5.2 Machine/Deep Learning-based Code Similarity Comparison

**Mono-Architecture Approaches.** Recent research has demonstrated the applicability of machine/deep learning techniques to code analysis (Han et al., 2017; Ding et al., 2019; Van Nguyen et al., 2017; Phan et al., 2017; Yan et al., 2019; Massarelli et al., 2019). Lee et al. propose Instruction2vec for converting assembly instructions to vector representations (Lee et al., 2017). PalmTree (Li et al., 2021) generates instruction embeddings by adopting the BERT model (Devlin et al., 2018). However, all these approaches work on a *single* ISA.

**Cross-Architecture Approaches.** A set of approaches target *cross-architecture* binary analysis (Feng et al., 2016; Xu et al., 2017; Chandramohan et al., 2016; Zuo et al., 2018; Redmond et al., 2019). Some exploit the code *statistical*. For example, Gemini (Xu et al., 2017) use manually selected features (e.g., the number of constants and function calls) to represent basic blocks, but ignore the meaning of instructions and dependency between them, resulting in significant information loss. InnerEye (Zuo et al., 2018) uses LSTM to encode each basic block into an embedding, but needs to train a *separate* model for each ISA.

For these approaches, in order to train their models, cross-architecture signals (i.e, labeled similar and dissimilar pairs of code samples across ISAs) are needed, which require a lot of engineering efforts. For example, InnerEye (Zuo et al., 2018) modifies the backends of various ISAs in the LLVM compiler (LLVM) to generate similar and dissimilar basic block pairs in different ISAs, while the dataset collection is identified as one of the challenges of this work. We instead build an unsupervised binary code translation model that does not require large parallel corpora. More importantly, our work can resolve the data scarcity problem in low-resource ISAs.

The concurrent work, UniMap (Wang et al., 2023), also aims to address the data scarcity issue in low-resource ISAs. We proposed entirely different approaches. UniMap learns cross-architecture instruction embeddings (CAIE) to achieve the goal. In contrast, our work takes a more progressive stride by translating code from low-resource ISAs to a high-resource ISA. As shown in Figure 2, UniMap stops at the CAIE learning stage, while our work moves beyond this point, concentrating on binary code translation. As we translate binary code to a high-resource ISA, our approach has several advantages. One advantage is that we can generate function embeddings by directly summing the CAIE of instructions within a function and use the function embeddings to measure similarity (as illustrated in the function similarly comparison task). This eliminates the need for a downstream model training, which however is required by UniMap. Another advantage is that we can directly use the existing downstream models that have already been trained on the high-resource ISA to test the translated code (as illustrated in the vulnerability discovery task). However, UniMap needs to retrain the existing models using the learned CAIE.

## 6 Discussion

Intermediate representation (IR) can be used to represent code of different ISAs (angr Documentation). Given two pieces of binaries from different ISAs, which have been compiled from the same piece of source code, even if they are converted into a common IR, the resulting IR code still looks quite different (see Figure 1 and 3 of (Pewny et al., 2015) and the discussion in (Wang et al., 2023)). As a result, existing works that leverage IR for analyzing binaries across ISAs have to perform further advanced analysis on the IR code (Pewny et al., 2015; David et al., 2017; Luo et al., 2019b, 2023).

## 7 Conclusion

Deep learning has been widely adopted for binary code analysis. However, the limited availability of data for low-resource ISAs hinders automated analysis. In this work, we proposed UNSUPERBIN-TRANS, which can leverage the easy availability of dataset in x86 to address the data scarcity problem in low-resource ISAs. We conducted experiments to evaluate the performance. For vulnerability discovery, UNSUPERBINTRANS is able to detect all vulnerable functions across ISAs. Thus, UNSU-PERBINTRANS offers a viable solution for addressing the data scarcity issue in low-resource ISAs.

NLP-inspired binary code analysis is a promising research direction, but not all NLP techniques are applicable to binary code analysis. Therefore, works like ours that identify and study effective NLP techniques for binary code analysis are valuable in advancing exploration along this direction. This work demonstrates how NLP techniques can be leveraged in binary code analysis and underscores the practical utility of such approaches.

## Limitations

In the evaluation, we consider two ISAs, x86 and ARM. There are many other ISAs, such as MIPS and PowerPC. We leave the evaluation on whether our approach works for other ISAs as future work.

In this work, we consider *instructions* as *words*, which is a natural choice and works well according to our evaluation. However, there exist other choices. For instance, we may also consider *raw bytes* (Liu et al., 2018; Guo et al., 2019; Raff et al., 2018) or *opcode* and *operands* (Ding et al., 2019; Li et al., 2021) as words. An intriguing research work is to study which choice works the best and whether it depends on ISAs and downstream tasks.

Given the large variety of binary analysis tasks and their complexity, we do not claim that our approach can be applied to all tasks. The fact that it works well for two important tasks (vulnerability discovery and code similarity detection). Demonstrating our approach for other applications is of great use. Much research can be done for exploring and expanding the boundaries of the approach.

## Ethics Statement

We state that this work does not violate any of the ethics and principles of scientific research in the field of computer science.

## Acknowledgments

This work was supported in part by the US National Science Foundation (NSF) under grants CNS-2304720. The authors would like to thank the anonymous reviewers for their valuable comments.

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
