# OpenReview forum: "Unsupervised Binary Code Translation with Application to Code Clone Detection and Vulnerability Discovery"
_EMNLP/2023/Conference — EMNLP 2023 Findings_

### Official Review · Reviewer_qjhC · 2023-08-04

**Soundness:** 3

**Excitement:**

3: Ambivalent: It has merits (e.g., it reports state-of-the-art results, the idea is nice), but there are key weaknesses (e.g., it describes incremental work), and it can significantly benefit from another round of revision. However, I won't object to accepting it if my co-reviewers champion it.

**Missing References:**

- Wang et al. Can a Deep Learning Model for One Architecture Be Used for Others? Retargeted-Architecture Binary Code Analysis.  Usenix 2023.

**Paper Topic And Main Contributions:**

The authors develop an approach to retargeting a deep neural network used for binary analysis on one ISA to work with others.  The issue is that certain esoteric or more rarely-used architectures have a scarce amount of well-labeled data that preclude training large, deep neural networks.  The authors develop cross-architectural instruction embeddings that abstract commonalities between high-resource ISAs like x86 and a lower-resource ISA like ARM or PPC.  The CAIEs are generated by mapping instruction embeddings of one program across multiple ISAs into the same embedding space.  Ultimately, the technique involves translating programs represented in one ISA to another ISA.

The technique translates a given low-resource ISA binary into another high-resource ISA (i.e., x86).  The translated binaries can then be used in subsequent downstream tasks.  The authors consider functional similarity comparison and vulnerability detection.  The authors provide relevant metrics (accuracy for functionality similarity comparison and F-1 for vulnerability detection).

**Questions For The Authors:**

- How does this approach compare against baselines with respect to your two downstream evaluations?
    - Author rebuttal claims to promise such evaluations.
- In what way is this an improvement over using intermediate representations like LLVM?  Especially considering that this paper depends on translating binaries to x86, is this not similar to converting to LLVM IR or similar?
    - partially addressed in the rebuttal
- In what way does this paper differ from the Usenix 2023 paper, "Can a Deep Learning Model for One Architecture Be Used for Others?
Retargeted-Architecture Binary Code Analysis" ?
    - The authors clarified this point.  It sounds like there is substantial overlap, but still novel contributions in the current submission.

**Reasons To Accept:**

- important problem.  Binary analysis is challenging, and representing binaries for deep neural networks is quite appropriate given the structure and difficulty in comprehending binary or assembly instructions -- it is a very natural application.
- cross architectural binary analysis is quite important given the increasing prevalence of embedded ISAs combined with the longstanding widespread availability of x86(-64)-based analyses.

**Reasons To Reject:**

- no comparison against baselines.  The functionality similarity comparison study reports only accuracy across optimization levels of binaries, but no baselines are considered.  This is a widely-understood binary analysis application and many papers have developed architecture-agnostic similarity comparison (or often reported as codesearch, which is a similar task).
    - rebuttal promises to add this evaluation
- in addition, the functionality similarity comparison methodology is questionable.  The authors use cosine similarity with respect to embeddings, which to me makes the experiment rather circular.  In contrast, I might have expected some type of dynamic analysis, testing, or some other reasoning to establish semantic similarity between code snippets.
    - rebuttal addresses this point.
- vulnerability discovery methodology is also questionable.  The authors consider a single vulnerability at a time, and while they acknowledge and address the data imbalance issue, I am not sure about the ecological validity of such a study.  Previous work has considered multiple CVEs or CWEs at a time, and report whether or not the code contains any such vulnerability.  Are the authors arguing that identifying one vulnerability at a time is an intended use case?  In any case, the results are difficult to interpret (or are marginal improvements at best).
    - addressed in rebuttal
- This paper is very similar to another accepted at Usenix 2023: Can a Deep Learning Model for One Architecture Be Used for Others?
Retargeted-Architecture Binary Code Analysis.  In comparison to that paper, I do not quite understand the novelty here, except perhaps for a slightly different evaluation/application domain.   I certainly acknowledge that this submission was made slightly before the Usenix 2023 proceedings were made available, but I would still want to understand how this differs given the overall similarity in idea (building embeddings that help a model target a new ISA).
     - addressed in rebuttal
- relatedly, only x86 and ARM appear to be considered in the evaluation (the authors discuss building datasets for these ISAs).  There are other ISAs to consider (e.g., PPC), and validating the approach against other ISAs would be important if claiming to build models that generalize to across architectures.
    - author rebuttal promises a followup evaluation

**Reproducibility:**

3: Could reproduce the results with some difficulty. The settings of parameters are underspecified or subjectively determined; the training/evaluation data are not widely available.

**Reviewer Confidence:**

4: Quite sure. I tried to check the important points carefully. It's unlikely, though conceivable, that I missed something that should affect my ratings.

---

> ### Author Rebuttal · Authors · 2023-08-27
>
> Thanks a lot for the insightful and valuable comments. Please find our response below.
>
> >  1. “*no comparison against baselines. The functionality similarity comparison study reports only accuracy across optimization levels of binaries, but no baselines are considered. This is a widely-understood binary analysis application and many papers have developed architecture-agnostic similarity comparison (or often reported as codesearch, which is a similar task).*”
>
> **Response**: Thanks for the suggestion. We will conduct a baseline comparison for the function similarity comparison task. Specifically, we will consider existing deep learning-based code similarity comparison models as the baseline methods, and train and test the models on ARM. The results of the baselines will be compared with those obtained by training the models on x86 and testing them on ARM (when testing on ARM, we will first use our model UNSUPERBINTRANS to translate ARM code to x86). Through this, we can evaluate the effectiveness of our approach’s translation capabilities: If the performance discrepancy is small, it indicates effective translation.
>
> > 2. “*in addition, the functionality similarity comparison methodology is questionable. The authors use cosine similarity with respect to embeddings, which to me makes the experiment rather circular. In contrast, I might have expected some type of dynamic analysis, testing, or some other reasoning to establish semantic similarity between code snippets.*”
>
> **Response**: We first translate code from ARM to x86 using our model UNSUPERBINTRANS. Through this, we have two pieces of code, both in x86. To measure the similarity between the two pieces of code, a complex way is to train a deep learning model. However, we opted for the simplest way, aiming to better demonstrate the good translation of our model. That is, we sum the instruction embeddings of all instructions within each function, and use the sum as the function embedding (this is similar to the way of computing sentence/document embeddings using word embeddings in NLP). Subsequently, the cosine similarity between the two function embeddings is computed to measure the function similarity. We believe this simplest way, compared to training a deep learning model (which can help capture code similarity), better highlights the good translation of our model.
>
> Thanks for the great suggestion. The process of translating assembly code from one ISA into the data-rich x86 ISA opens the door to employing both static and dynamic analysis techniques for code similarity comparison. In this work, we focus on static analysis techniques, wherein we employ deep learning-based models to analyze the code. While dynamic analysis techniques represent a valid direction, they are beyond the scope of this work.
>
> > 3. “*vulnerability discovery methodology is also questionable. The authors consider a single vulnerability at a time, and while they acknowledge and address the data imbalance issue, I am not sure about the ecological validity of such a study. Previous work has considered multiple CVEs or CWEs at a time, and report whether or not the code contains any such vulnerability. Are the authors arguing that identifying one vulnerability at a time is an intended use case? In any case, the results are difficult to interpret (or are marginal improvements at best).*”
>
> **Response**: For the vulnerability discovery task, there are two common handlings in the literature. The first, as mentioned by the reviewer, considers multiple CVEs or CWEs at a time. The second one, which we adopted, considers one CVE at a time, such as Gemini (Xu et al., 2017).
>
> We want to clarify that our interpretation of the results does not focus on performance improvements; rather, our focus lies on performance discrepancies, specifically performance decreases. When comparing our model's results to the baseline, we consider our model's translation capabilities to be effective if the observed decrease is minimal. For the details, please refer to our response to the Question numbered 6 (below).
>
> > 4. “*This paper is very similar to another accepted at Usenix 2023: Can a Deep Learning Model for One Architecture Be Used for Others? Retargeted-Architecture Binary Code Analysis. In comparison to that paper, I do not quite understand the novelty here, except perhaps for a slightly different evaluation/application domain. I certainly acknowledge that this submission was made slightly before the Usenix 2023 proceedings were made available, but I would still want to understand how this differs given the overall similarity in idea (building embeddings that help a model target a new ISA).*”
>
> **Response**: The USENIX 2023 work focuses on the same goal as ours (i.e., addressing the data scarcity issue in low-resource ISAs). However, we proposed entirely different approaches. The USENIX work learns cross-architecture instruction embeddings (CAIE) to achieve the goal. In contrast, our work takes a more progressive stride by translating code in low-resource ISAs to a data-rich ISA. A visual reference to these distinctions can be found in Figure 2, where the USENIX work stops at the CAIE learning stage, while our work moves beyond this point, concentrating on binary code translation.
>
> As we translate binary code to a data-rich ISA, our approach has several advantages. One advantage is that we can directly use the existing models that have already been trained on the data-rich ISA to test the translated code (converted into the data-rich ISA by our model UNSUPERBINTRANS). However, the USENIX work needs to **retrain** the existing models using their learned CAIE for analyzing code in different ISAs. Another advantage is that our approach opens the door to utilize existing models, which are designed to analyze code for extracting features from binaries, to analyze the translated code. However, the USENIX work lacks support of this potential direction.
>
> > 5. “relatedly, only x86 and ARM appear to be considered in the evaluation (the authors discuss building datasets for these ISAs). There are other ISAs to consider (e.g., PPC), and validating the approach against other ISAs would be important if claiming to build models that generalize to across architectures.”
>
> **Response**: Thanks for the suggestion! We will include other ISAs (e.g., PPC) into our evaluation, and conduct the downstream tasks in terms of these ISAs.
>
> > 6. “How does this approach compare against baselines with respect to your two downstream evaluations?”
>
> **Response**: Our work aims to address the data scarcity issue in low-resource ISAs for binary code analysis. To achieve this, we introduce UNSUPERBINTRANS, a model trained to translate assembly code in one ISA to assembly code in the data-rich ISA x86. Through the application of UNSUPERBINTRANS, we can resolve the data scarcity issue. Specifically, given a downstream binary analysis task, we can use the downstream model trained on x86 to test binary code in another ISA by translating the code to x86.
>
> To demonstrate the translation effectiveness of our approach, we conducted a baseline comparison, wherein we trained a downstream model on ARM and tested the trained model on ARM, without employing any translation. As one would anticipate, the model trained and tested on ARM data should outperform a model trained on x86 and tested on translated ARM data.
>
> If the performance discrepancy is small, it signifies effective translation. Our results in Table 4 show that our model consistently demonstrates close proximity in most cases when comparing to the baseline. Thus, our approach has good translation capabilities.
>
> > 7. “In what way is this an improvement over using intermediate representations like LLVM? Especially considering that this paper depends on translating binaries to x86, is this not similar to converting to LLVM IR or similar?”
>
> **Response**: Our approach differs from the process of converting binaries to LLVM IR. Given two pieces of binaries from different ISAs, which have been compiled from the same piece of source code, even if they are converted into a common IR, the resulting IR code still looks quite different. Please see examples shown in Figure 1 and Figure 3 of the work (Pewny et al., 2015). Moreover, the recent USENIX 2023 work also discusses IR in Section 4.3.1. Thus, binary code similarity (which cannot be addressed through code lifting to IR) has motivated intensive studies.

---

### Official Review · Reviewer_SuGi · 2023-08-05

**Soundness:** 4

**Excitement:**

4: Strong: This paper deepens the understanding of some phenomenon or lowers the barriers to an existing research direction.

**Paper Topic And Main Contributions:**

The authors in this work present a novel task of binary translation and a dataset for its evaluation. The authors also use combination of pre-existent tools to create a baseline for the task.


**Reasons To Accept:**

A novel task of binary translation and a novel dataset for evaluation in this task.


**Reasons To Reject:**

The claimed novel model is not actually novel as it is a straight combination of already existing tools.

**Reproducibility:**

4: Could mostly reproduce the results, but there may be some variation because of sample variance or minor variations in their interpretation of the protocol or method.

**Reviewer Confidence:**

4: Quite sure. I tried to check the important points carefully. It's unlikely, though conceivable, that I missed something that should affect my ratings.

---

> ### Author Rebuttal · Authors · 2023-08-26
>
> Thanks a lot for the insightful and valuable comments. Please find our response below.
>
> > “*The claimed novel model is not actually novel as it is a straight combination of already existing tools.*”
>
> **Response:** We will tone down the claim and properly describe the novelty of this work. Our work is the first to explore the translation of binary code between different ISAs. We leverage the ideas and techniques in NMT to a quite different area, binary code analysis. NLP-inspired binary code analysis is a promising research direction, but not all NLP techniques are applicable to binary code analysis. Therefore, works like ours that identify and study effective NLP techniques for binary code analysis are valuable in advancing exploration along this direction. We adapt NMT techniques to binary code analysis, aiming to address the data scarcity issue in low-resource ISAs.  Our evaluation shows how the adaptation works and why it is useful through two downstream tasks, function similarity comparison and vulnerability discovery.

---

### Official Review · Reviewer_6b4d · 2023-08-05

**Soundness:** 2

**Excitement:**

2: Mediocre: This paper makes marginal contributions (vs non-contemporaneous work), so I would rather not see it in the conference.

**Paper Topic And Main Contributions:**

The authors use the ideas and techniques of neural machine translation to translate binaries in low-resource architectures into binaries in high-resource architectures. They show the quality of the translation through two downstream tasks: code similarity detection and vulnerability discovery.


**Reasons To Accept:**

The concept and framework of translating low-resource binaries to high-resource binaries is a strength of this paper.


**Reasons To Reject:**

1) While the primary contribution is about the translation of low-resource architecture code, the paper lacks detailed information and analysis of these results. Section 5.2 only presents BLEU scores without any in-depth analysis or explanation. The reported scores don't seem to vary significantly across different optimizations and settings, which raises questions about how the model converged, among other things. These aspects need more elaboration.
2) The code similarity and vulnerability discovery task seem simplistic. The definition of similarity is too dependent on same name and derived from same source code. What about the case when two functions have different names but they do the same task. No significant differences are discernible compared to the baseline models.
3) There have been previous studies on intermediate representation-based binary analysis and rewriting systems. It's essential to highlight how your approach is different or improved compared to existing methods. I found no comparisons in your paper.
4) The paper should elaborate on how this model can be generalized to handle new coding styles, functions, and more. It would be helpful if you could elaborate on the model's scalability and adaptability.


**Reproducibility:**

3: Could reproduce the results with some difficulty. The settings of parameters are underspecified or subjectively determined; the training/evaluation data are not widely available.

**Reviewer Confidence:**

3: Pretty sure, but there's a chance I missed something. Although I have a good feel for this area in general, I did not carefully check the paper's details, e.g., the math, experimental design, or novelty.

**Typos Grammar Style And Presentation Improvements:**

The paper is clear overall.

Authors should add an introduction or overview before the first sub-sections in sections 2 and 4.

---

> ### Author Rebuttal · Authors · 2023-08-26
>
> Thanks a lot for the insightful and valuable comments. Please find our response below.
>
> > 1. “*While the primary contribution is about the translation of low-resource architecture code, the paper lacks detailed information and analysis of these results. Section 5.2 only presents BLEU scores without any in-depth analysis or explanation. The reported scores don't seem to vary significantly across different optimizations and settings, which raises questions about how the model converged, among other things. These aspects need more elaboration.*”
>
> **Response**: In Section 5.2, we presented the BLEU scores when translating ARM to x86 in terms of four different optimization levels. The BLEU scores are around 0.76 and 0.77, which shows the good translation performance of our model (given that BLEU scores range from 0 to 1, with higher scores indicating better translation quality).
>
> We then used a few randomly selected samples as examples to illustrate the robust translation capability of our model. We are open to further analysis and explanations. Please kindly apprise us of your thoughts, and we would be more than willing to include them into the revision.
>
> We find ourselves uncertain regarding your question: “*The reported scores don't seem to vary significantly across different optimizations and settings, which raises questions about how the model converged, among other things.*” Our perspective contrasts here, as we observe minimal score variance, showing the convergence of our model.
>
> > 2. “*The code similarity and vulnerability discovery task seem simplistic. The definition of similarity is too dependent on same name and derived from same source code. What about the case when two functions have different names but they do the same task. No significant differences are discernible compared to the baseline models.*”
>
> **Response**: Besides function names, we used a method (n-gram) similar to that in InnerEye (Zuo et al., 2019) (see Section V.D.2) to double check whether two functions are indeed dissimilar.
>
> **No significant differences between our model and the baseline model are what we expected, which show the good translation capability of our approach**. Our work aims to address the data scarcity issue in low-resource ISAs for binary code analysis. To achieve this, we introduce UNSUPERBINTRANS, a model trained to translate assembly code in one ISA to assembly code in the data-rich ISA x86. Through the application of UNSUPERBINTRANS, we can resolve the data scarcity issue. Specifically, given a downstream binary analysis task, we can use the downstream model trained on x86 to test binary code in another ISA by translating the code to x86.
>
> To demonstrate the translation effectiveness of our approach, we conducted a baseline comparison, wherein we trained a downstream model on ARM and tested the trained model on ARM, without employing any translation. As one would anticipate, the model trained and tested on ARM data should outperform a model trained on x86 and tested on translated ARM data.
>
> If the performance discrepancy is small, it signifies effective translation. Our results in Table 4 show that our model consistently demonstrates close proximity in most cases when comparing to the baseline. Thus, our approach has good translation capabilities.
>
> > 3. “*There have been previous studies on intermediate representation-based binary analysis and rewriting systems. It's essential to highlight how your approach is different or improved compared to existing methods. I found no comparisons in your paper.*”
>
> **Response**: Existing studies that leverage intermediate representation (IR) cannot resolve the data scarcity issue in low-resource ISAs. First, given two pieces of binaries from different ISAs, which have been compiled from the same piece of source code, even if they are converted into a common IR, the resulting IR code still looks quite different. As a result, existing works that leverage IR for analyzing binaries across ISAs have to perform further advanced analysis (e.g., training deep learning models) on the IR code.
>
> Second, for a given downstream task, existing works require the task-specific data from all ISAs considered to train their downstream models. In contrast, for a given downstream task, our approach only needs the task-specific data in a data-rich ISA (such as x86) to train a downstream model on x86. Then, when confronted with the need to analyze code in a low-resource ISA, we can employ our model UNSUPERBINTRANS to translate the code to x86, and use the downstream model (trained on x86) to test the translated code.
>
> It should be highlighted that the training of UNSUPERBINTRANS only needs unlabelled data, which can be obtained conveniently via cross-compiling some source code. This is similar to the NLP works that utilize machine translation to deal with the data scarcity issue in low-resource languages for various NLP tasks such as cross-lingual document classification.
>
> We will include the discussion in the revision of this paper. Moreover, existing works based on intermediate representation are also discussed in Section 4.3.1 in the recently published paper:
>
> * Wang, J., Sharp, M., Wu, C., Zeng, Q. and Luo, L., 2023. Can a Deep Learning Model for One Architecture Be Used for Others? Retargeted-Architecture Binary Code Analysis. In 32nd USENIX Security Symposium (USENIX Security 23) (pp. 7339-7356).
>
> > 4. “The paper should elaborate on how this model can be generalized to handle new coding styles, functions, and more. It would be helpful if you could elaborate on the model's scalability and adaptability.”
>
> **Response**: Great suggestion! We acknowledge that the models we trained are unlikely to generalize to all kinds of code. That is an ambitious goal, and the community is not there yet, although our work can be considered as a good effort towards it; that is, the generalizability across different architectures (for certain Linux code).
>
> We agree that the ambitious goal can be attained in one shot. To mitigate the limitation, we may increase the diversity of the dataset by involving more kinds of code (with new coding styles, functions, and more). We will elaborate this in the revision of this paper.

---

### Meta-Review · Area_Chair_Uf8m · 2023-09-19

**Recommendation:** 2

**Metareview:**

The reviewers agree that the main strength of the paper is the proposed conceptual framework for translating binaries, and the dataset created to evaluate it. That said, the paper does not compare against existing baselines for the function similarity comparison task. The paper over-claims some of the contributions and novelty, but based on rebuttal response, the authors intend to change it in the revision, including clearly contrasting the work and highlighting novelty over the Usenix 2023 paper.

---

### Decision · Program_Chairs · 2023-10-07

**Decision:**

Accept-Findings

**Comment:**

The reviewers agree that the main strength of the paper is the proposed conceptual framework for translating binaries, and the dataset created to evaluate it. That said, the paper does not compare against existing baselines for the function similarity comparison task. The paper over-claims some of the contributions and novelty, but based on rebuttal response, the authors intend to change it in the revision, including clearly contrasting the work and highlighting novelty over the Usenix 2023 paper.